

# Genomic analysis of ST88 community-acquired methicillin resistant *Staphylococcus aureus* in Ghana

Grace Kpeli[1,2,3], Andrew H. Buultjens[4], Stefano Giulieri[4], Evelyn Owusu-Mireku[1], Samuel Y. Aboagye[1], Sarah L. Baines[4], Torsten Seemann[4,5], Dieter Bulach[4,5], Anders Gonçalves da Silva[4], Ian R. Monk[4], Benjamin P. Howden[4,6,7], Gerd Pluschke[2,3], Dorothy Yeboah-Manu[1] and Timothy Stinear[4]

[1] Department of Bacteriology, Noguchi Memorial Institute for Medical Research, University of Ghana, Accra, Ghana
[2] Department of Molecular Parasitology and Immunology, Swiss Tropical and Public Health Institute, Basel, Switzerland
[3] University of Basel, Basel, Switzerland
[4] Department of Microbiology and Immunology, Doherty Applied Microbial Genomics, Doherty Institute for Infection and Immunity, University of Melbourne, Melbourne, VIC, Australia
[5] University of Melbourne, Victorian Life Sciences Computation Initiative, Melbourne, VIC, Australia
[6] Department of Microbiology and Immunology, Microbiological Diagnostic Unit Public Health Laboratory, Doherty Institute for Infection & Immunity, University of Melbourne, Melbourne, VIC, Australia
[7] Department of Infectious Diseases, Austin Health, Heidelberg, VIC, Australia

Corresponding authors
Grace Kpeli,
gkpeli@noguchi.ug.edu.gh
Timothy Stinear,
tstinear@unimelb.edu.au

## ABSTRACT

**Background:** The emergence and evolution of community-acquired methicillin resistant *Staphylococcus aureus* (CA-MRSA) strains in Africa is poorly understood. However, one particular MRSA lineage called ST88, appears to be rapidly establishing itself as an "African" CA-MRSA clone. In this study, we employed whole genome sequencing to provide more information on the genetic background of ST88 CA-MRSA isolates from Ghana and to describe in detail ST88 CA-MRSA isolates in comparison with other MRSA lineages worldwide.

**Methods:** We first established a complete ST88 reference genome (AUS0325) using PacBio SMRT sequencing. We then used comparative genomics to assess relatedness among 17 ST88 CA-MRSA isolates recovered from patients attending Buruli ulcer treatment centres in Ghana, three non-African ST88s and 15 other MRSA lineages.

**Results:** We show that Ghanaian ST88 forms a discrete MRSA lineage (harbouring SCC*mec*-IV [2B]). Gene content analysis identified five distinct genomic regions enriched among ST88 isolates compared with the other *S. aureus* lineages. The Ghanaian ST88 isolates had only 658 core genome SNPs and there was no correlation between phylogeny and geography, suggesting the recent spread of this clone. The lineage was also resistant to multiple classes of antibiotics including β-lactams, tetracycline and chloramphenicol.

**Discussion:** This study reveals that *S. aureus* ST88-IV is a recently emerging and rapidly spreading CA-MRSA clone in Ghana. The study highlights the capacity of

small snapshot genomic studies to provide actionable public health information in resource limited settings. To our knowledge this is the first genomic assessment of the ST88 CA-MRSA clone.

## INTRODUCTION

Since the 1990s, community-acquired methicillin-resistant *Staphylococcus aureus* (CA-MRSA) infections have been increasing worldwide (*Centers for Disease Control and Prevention, 2013*; *de Kraker et al., 2011*). CA-MRSA clones are known to be more virulent than hospital-acquired MRSA, with infections linked to significant mortality and morbidity (*Chambers, 2001*; *Chua et al., 2011*, *2014*; *Etienne, 2005*; *Kourbatova et al., 2005*; *Seybold et al., 2006*). First reported in Australia and the United States, CA-MRSA occurrence has been increasing, with epidemics due to clones such as ST8 USA300 in the United States (*Diekema et al., 2014*), ST93 and ST1 in Australia (*Coombs et al., 2009*), ST80 in Europe (*Otter & French, 2010*), ST59 in China and Taiwan (*Chen & Huang, 2014*), ST772 in India (*D'Souza, Rodrigues & Mehta, 2010*; *DeLeo et al., 2010*; *Nadig et al., 2012*; *Shambat et al., 2012*) and ST72 in South Korea (*Kim et al., 2007*). Other identified CA-MRSA clones belong to ST30 (South West Pacific clone) (*Williamson, Coombs & Nimmo, 2014*), ST45 (Berlin clone) (*Witte et al., 1997*), ST1 (USA400) (*DeLeo et al., 2010*) and ST78 (Western Australian MRSA-2) (*Williamson, Coombs & Nimmo, 2014*). In Africa, the distribution of MRSA clones in general is not well understood (*Abdulgader et al., 2015*). A recent review on MRSA in Africa with data from 15 of the 54 countries identified community clones of ST8-IV [2B] (USA300) and ST88-IV [2B] "West Australia MRSA-2 clone" in both community and health care associated infections in seven countries and a "Brazilian/Hungarian clone" ST239-III [3A] in hospital acquired infections in nine countries (*Abdulgader et al., 2015*). The European ST80-IV [2B] clone was limited to Algeria, Egypt and Tunisia while clonal types ST22-IV [2B], ST36-II [2A] and ST612-IV [2B] have only been reported so far in South Africa (*Abdulgader et al., 2015*). Among the two CA-MRSA clones, the ST8IV [2B] clone is an internationally disseminated clone recognized in every continent except Antarctica (*David & Daum, 2010*). The ST88-IV [2B] CA-MRSA clone however is predominant in Sub-Saharan Africa (West, Central and East Africa) with reported rates of 24.2–83.3% of all MRSA isolates (*Schaumburg et al., 2014*). Studies from Angola (*Conceição et al., 2014*), Cameroon (*Breurec et al., 2011*), Gabon (*Ngoa et al., 2012*; *Schaumburg et al., 2011*), Ghana (*Amissah et al., 2015*; *Egyir et al., 2013*, *2014*), Madagascar (*Breurec et al., 2011*), Niger (*Breurec et al., 2011*), Nigeria (*Ghebremedhin et al., 2009*; *Raji et al., 2013*; *Shittu et al., 2012*) and Senegal (*Breurec et al., 2011*) have identified it as a major circulating clone within both hospital and community settings. It was also detected in children from West Africa who underwent surgery in Switzerland but had been hospitalized in their home countries prior to surgical treatment

(*Blanc et al., 2007*). Globally, this clone has been identified in China (*Yu et al., 2008*) and Japan (*Maeda et al., 2012*) in lower proportion (5.3–12.5%) than in Africa and sporadically in Belgium (*Denis et al., 2005*), Portugal (*Aires-de-Sousa et al., 2008*) and Sweden (*Fang et al., 2008*).

Control of MRSA infections is assisted by a thorough knowledge of the epidemiology and dissemination of specific clones. To this end, we employed whole genome sequencing and comparative genomics to describe in detail ST88 CA-MRSA isolates in comparison to other MRSA lineages worldwide.

## MATERIALS AND METHODS

### Bacterial isolates and antibiogram analysis

The 17 ST88 *S. aureus* isolates analysed from Ghana are listed in Table 1 and comprised five strains isolated in the Akwapim South District (Eastern Region) of Ghana with previously published genome data (GenBank accession numbers: LFNJ00000000, LFNI00000000, LFNH00000000, LFMH00000000, LFMG00000000) (*Amissah et al., 2015*) and 12 isolates recovered from wounds of 11 patients attending Buruli ulcer (BU) treatment centres in the Ga-South and Ga-West municipalities (Greater Accra Region) of Ghana with two isolates from one patient; one a PVL positive isolate and the other PVL negative. Patients were outpatients, nine of whom had laboratory confirmed BU. Initial isolate identification was made using colony and microscopic morphology, catalase and coagulase biochemical reactions and a Staphylase kit BD BBL™ Staphyloslide Latex Test (Becton, Dickinson and Company, NJ, USA) for further confirmation. Antibiograms were determined using the Kirby Bauer disc diffusion method according to CLSI guidelines (*CLSI, 2014*) and PCR targeting the *mecA* gene (*Oliveira & de Lencastre, 2002*) for identification of MRSA. Ethical clearance was obtained from the institutional review board of the Noguchi Memorial Institute for Medical Research (NMIMR) (Federal-wide Assurance number FWA00001824). All study participants were well informed of the study objectives and written informed consent was obtained either from the patient or from the guardian of the patient.

### DNA extraction, whole genome sequencing and analysis

Genomic DNA was extracted from isolates using the Qiagen DNeasy kit and protocol (Qiagen, Hilden, Germany). DNA libraries were prepared using Nextera XT (Illumina, San Diego, CA, USA) and whole genome sequencing was performed using the Illumina MiSeq with $2 \times 300$ bp chemistry. Small Molecular Real Time sequencing was performed on the RS-II (Pacific Biosciences, California, United States) using P6-C4 chemistry, and reference genome assembly was performed as described (*Baines et al., 2016*).

### Read mapping, variant calling and phylogenomic analysis

The sequence reads were processed using *Nullarbor* (nullarbor.pl 0.6, https://github.com/tseemann/nullarbor), a recently developed bioinformatics pipeline for public health microbial genomics as described previously (*Kwong et al., 2016*). *S. aureus* ST88 raw

**Table 1  _S. aureus_ ST88 isolates tested in this study.**

| Isolate ID | Origin (Ghana) | Phenotypic resistance | Genotype (_spa_, _agr_, PVL) | Reference |
|---|---|---|---|---|
| Sa_NOG-W02 | Greater Accra Region | cld, tet, amp, ery, fox, ctx, chl, cro | t939, agr-3, PVL+ | This study |
| Sa_NOG-W25 | Greater Accra Region | gen, amk, cld, str, amp, tet, sxt, cfx, ctx, chl, cro | t448, agr-3, PVL− | This study |
| Sa_NOG-W11 | Greater Accra Region | str, amk, gen, sxt, cfx, cld, fox, ctx, tet, chl, cro, amp, ery | t186, agr-3, PVL+ | This study |
| Sa_NOG-W13 | Greater Accra Region | gen, str, amk, ctx, tet, chl, cro, sxt, cfx, amp, cld, fox | 07-12-12-118-13-13, agr-3, PVL+ | This study |
| Sa_NOG-W01 | Greater Accra Region | amk, cfx, tet, ctx, chl, cro, fox | t186, agr-3, PVL+ | This study |
| Sa_NOG-W10 | Greater Accra Region | sxt, ery, gen, str, amk, cld, amp, cfx, tet, fox, ctx, chl, cro | t186, agr-3, PVL− | This study |
| Sa_NOG-W07 | Greater Accra Region | gen, str, amp, tet, sxt, cfx, chl, cro, ctx, fox, cld, ery, | t448, agr-3, PVL− | This study |
| Sa_NOG-W14 | Greater Accra Region | gen, ery, sxt, amk, cld, str, tet, amp, cfx, ctx, chl, cro, fox, | t2649, agr-3, PVL+ | This study |
| Sa_NOG-W04 | Greater Accra Region | sxt, ery, gen, str, amk, amp, cfx, tet, fox, ctx, chl, cro | 07-12-21-17-13-13-34-34-33-34-34, agr-3, PVL− | This study |
| Sa_NOG-W06 | Greater Accra Region | sxt, gen, amk, cld, amp, tet, cfx, fox, chl, cro | t786, agr-3, PVL− | This study |
| Sa_NOG-W24 | Greater Accra Region | gen, sxt, amk, str, amp, tet, cfx, ctx, chl, cro, fox, | t786, agr-3, PVL+ | This study |
| Sa_NOG-W05 | Greater Accra Region | ery, amk, str, amp, cfx, tet, sxt, cld, | t186, agr-3, PVL− | This study |
| BU_G0701_t5 | Eastern Region | fox, ben, oxa, tet, chl | t786, agr-3, PVL− | _Amissah et al. (2015)_ |
| BU_G0201_t8 | Eastern Region | fox, ben, oxa, tet, chl | t786, agr-3, PVL− | _Amissah et al. (2015)_ |
| BU_G0202_t2 | Eastern Region | fox, ben, oxa, tet, chl | t786, agr-3, PVL− | _Amissah et al. (2015)_ |
| BU_G1905_t3 | Eastern Region | fox, ben, oxa, tet, chl | t786, agr-3, PVL− | _Amissah et al. (2015)_ |
| BU_W13_11 | Eastern Region | fox, ben, oxa, tet, chl | t186, agr-3, PVL− | _Amissah et al. (2015)_ |

**Notes:**
oxa, Oxacillin; fox, cefoxitin; tet, tetracycline; chl, chloramphenicol; cfx, cefuroxime; ery, erythromycin; cld, clindamycin; sxt, sulphamethxazole-trimethoprim; amk, amikacin; str, streptomycin; amp, ampicillin; ctx, cefotaxime; cro, ceftriaxone; gen, gentamicin; ben, benzylpenicillin; _spa, Staphylococcus aureus_; Protein A, _agr_, Accesory Gene regulator; PVL, Pantone Valentine Leukocidin toxin.

sequence reads with accession numbers ERS1354589-600 have been deposited in the European Nucleotide Archive (ENA), Project PRJEB15428. Ortholog clustering was performed using Roary (_Page et al., 2015_) and was visualized with Fripan (http://drpowell. github.io/FriPan/). Recombination within the core genome was inferred using ClonalFrameML v1.7 (_Didelot & Wilson, 2015_) with the whole genome alignment generated by Nullarbor. Using FastTree v2.1.8 (_Price, Dehal & Arkin, 2010_), a ML tree was generated and used as a guide tree for ClonalFrameML. Positions in the reference genome that were not present in at least one genome (non-core) were omitted from the analysis using the "ignore_incomplete_sitestrue" option and providing ClonalFrameML with a list of all non-core positions. Maximum likelihood trees were constructed using a recombination free SNP alignment using FastTree. Bootstrap support was derived from comparisons between the original trees against 1,000 trees that were built upon pseudo-alignments (sampled from the original alignment with replacement).

## RESULTS AND DISCUSSION

### ST88 complete reference genome

A prerequisite for high-resolution comparative genomics by read-mapping is a high-quality, complete reference genome, closely related to the bacterial population under investigation (*Kwong et al., 2016*). There were no fully assembled ST88 *S. aureus* genomes publicly available.

Hence, to address this issue, we selected the methicillin-susceptible, penicillin-resistant ST88 *S. aureus* isolate AUS0325. This clinical isolate was obtained in 2013 from a patient in Melbourne, Australia who had a persistent infection of a prosthetic joint, and was part of a separate, unpublished study. The AUS0325 genome comprised a 2,771,577 bp circular chromosome with 32.9% GC content. There were no plasmids; the beta-lactamase operon (*bla*) was carried by the Tn*552* transposon and integrated into the chromosome. The overall chromosome architecture of AUS0325 was like representative *S. aureus* genomes from other community-associated lineages (ST1, ST8 and ST93) but with five distinct regions of difference, discussed in more detail below (Fig. 1A). We took advantage of the PacBio data to define the Sa_aus0325 methylome. Motif analysis and inspection of the AUS0325 annotation identified two active type I restriction modification *hsdMS* loci. Protein alignment of the two *hsdS* alleles with previously characterised *hsdS* proteins allowed the attribution of target recognition sequences to either allele (*Monk et al., 2015*) (Table 2). The first *hsdS* recognized a motif not previously described, while the second *hsdS* contained an identical sequence to the target recognition domain-2 of CC30-2 and ST93-2, which recognises TCG (Table 2).

### ST88 population structure

To understand the genomic diversity and evolutionary origin of the ST88 isolates, a core genome phylogeny was inferred by mapping reads from the 17 ST88 isolates (Table 1; Fig. 1B), two published ST88 MRSA genomes from Lebanon and USA and 15 other geographically and genetically distinct *S. aureus* clones to AUS0325 (Table 3; Fig. 1B). To assess the clonal ancestry, SNPs within inferred regions of recombination (71,862 clonal SNPs; 26,570 recombinogenic SNPs) (Fig. S1) were removed and a maximum likelihood phylogenomic tree was established using the clonal core SNP alignment (71,862 SNPs). All 20 ST88 genomes formed a discrete, closely related lineage, defined by only 1,759 core genome SNPs, compared with 71,862 SNPs among all 35 *S. aureus* genomes (Figs. 1B and 2A). The global tree was rooted using *Bacillus_subtilis*_B4068 (GenBank ID: JXHK00000000) (*Berendsen et al., 2016*) as an out-group and this phylogeny indicated ST88 shares a most recent common ancestor (MRCA) with ST72 (Fig. 1A).

Five distinct genomic regions were identified by ortholog comparisons, enriched among the ST88 genomes compared to the 15 other diverse *S. aureus* genomes. These regions included νSAα (GI-3, Fig. 1) that harboured 10 staphylococcal superantigen-like (*ssl*) genes, of which four were uniquely present in the ST88 isolates. Upregulation of SSLs has been reported in some CA-MRSA strains and may be involved in neutrophil and complement activation (*Foster, 2005*; *Voyich et al., 2005*). GI-3 also harboured the first

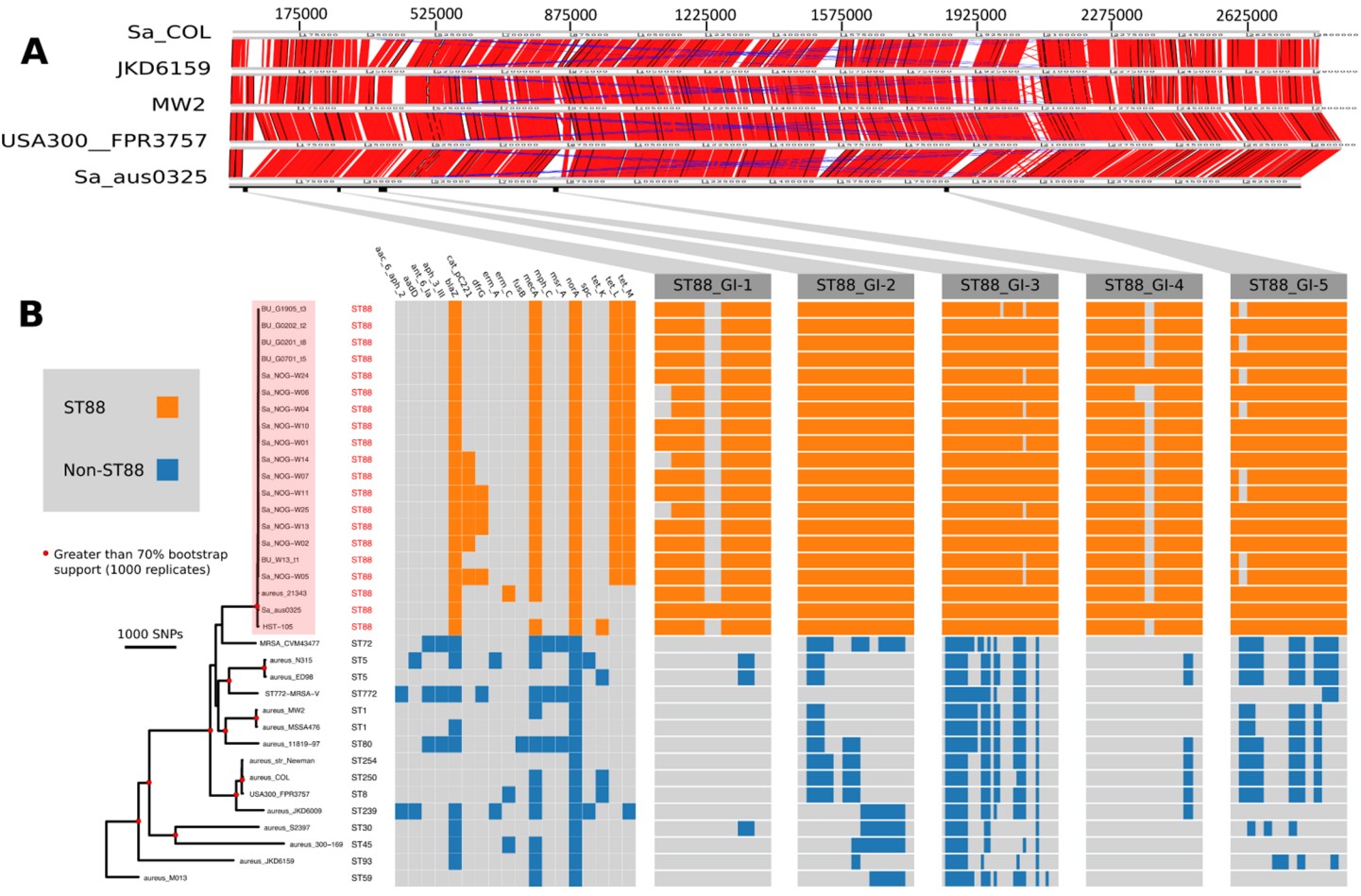

**Figure 1** **Comparative genomic analysis of *S. aureus* ST88.** (A) DNA–DNA comparisons visualized using the Artemis comparison tool of three CA-MRSA representative chromosomes and *S. aureus* COL against the complete chromosome of ST88 isolate AUS0325. (B) Core genome phylogeny and accessory genome elements identified among ST88 isolates. The phylogeny was based on an alignment of 71,862 non-recombinogenic core genome SNPs (indels excluded) and inferred using FastTree. Nodes with greater than 70% bootstrap support (1,000 replicates) are labelled with red dots. Antibiotic resistance genes were identified using Abricate (https://github.com/tseemann/abricate) and genomic islands (GIs) enriched among ST88 isolates were identified by ortholog comparisons using Roary and visualized using FriPan. CDS present in specific GI are listed in Table S1.

of the two functional type I restriction modification *hsdMS* loci (see above, Table 2). GI-1 and GI-4 may be mobile integrative elements of unknown function with the presence of putative integrases and 4 and 12 CDS, respectively, all encoding hypothetical proteins. GI-1 also harbours elements of a putative restriction modification system (Fig. 1; Table S1). GI-2 contains 13 CDS. Most of unknown function although three CDS may encode membrane proteins (Fig. 1; Table S1). GI-5 had 14 CDS that included the second of the type I restriction modification *hsdMS* loci and seven CDS encoding putative proteases (Table 2; Table S1).

## Evolution and molecular epidemiology of ST88 in Ghana

To assess the evolutionary relationships among the ST88 genomes, a phylogenomic tree comprised exclusively of ST88 genomes was established using clonal, core SNPs

**Table 2  Sa_aus0325 methylome analysis.**

| HsdS (nucleotide position) | TRD1 | N | TRD2 |
|---|---|---|---|
| 397,724 ≥ 399,280 | ACC | 5 | RTGT |
| 1,849,852 ≤ 1,851,408 | GAG | 6 | TCG |

**Table 3  Comparator reference genomes.**

| Sequence type | Region/country of origin | MSSA/MRSA | Reference strain | Assembly/ accession number |
|---|---|---|---|---|
| ST8 | USA/Canada | CA-MRSA | *Staphylococcus aureus* subsp. *aureus* USA 300 FPR 3757 | NC_007793.1 |
| ST 1 | USA/Canada | CA-MRSA | *Staphylococcus aureus* subsp. *aureus* MW2 | NC_003923.1 |
| ST 80 | Europe | CA-MRSA | *Staphylococcus aureus* 11819-97 | NC_017351.1 |
| ST45 | Europe/USA/ Canada | CA-MRSA | *Staphylococcus aureus* subsp. *aureus* 300-169 | GCA_000534855.1 |
| ST 30 | Europe/ Australia/Asia | CA-MRSA | *Staphylococcus aureus* subsp. *aureus*_S2397 | GCA_000577595.1 |
| ST 72 | Asia | CA-MRSA | *Staphylococcus aureus* MRSA_CVM43477 | GCA_000830555.1 |
| ST 59 | Asia | CA-MRSA | *Staphylococcus aureus* subsp. *aureus* M013 | NC_016928.1 |
| ST93 | Australia | CA-MRSA | *Staphylococcus aureus* subsp. *aureus* JKD 6159 | NC_017338.1 |
| ST 250 | England | HA-MRSA | *Staphylococcus aureus* subsp. *aureus* COL | NC_002951.2 |
| ST254 | Japan | MSSA | *Staphylococcus aureus* subsp. *aureus* Newman | NC_009641.1 |
| ST1 | United Kingdom | MSSA | *Staphylococcus aureus* subsp. *aureus* MSSA476 | NC_002953.3 |
| ST5 | Ireland | MSSA | *Staphylococcus aureus* subsp. *aureus* ED98 | NC_013450.1 |
| ST5 | Japan | MRSA | *Staphylococcus aureus* subsp. *aureus* N315 | NC_002745.2 |
| ST 239 | Australia | MRSA | *Staphylococcus aureus* subsp. *aureus* JKD 6008 | NC_017341.1 |
| ST772 | India | MRSA | *Staphylococcus aureus* subsp. *aureus*_ST772-MRSA | GCA_000516935.1 |
| ST 88 | Lebanon | MRSA | HST-105 | GCA_000564895.1 |
| ST 88 | United States | MSSA | *Staphylococcus aureus* subsp. *aureus*_21343 | GCA_000245595.2 |

(1,759 clonal SNPs; 207 recombinogenic SNPs) (Fig. S2; Fig. 2A). The tree was rooted using an ST93 genome (Sa_JKD6159) as an out-group. The phylogeny and the restricted genomic diversity (658 core SNPs) suggests that the spread of ST88 MRSA in Ghana is a recent phenomenon, with isolates from the United States, Australia and Lebanon ancestral to the spread of these isolates in Ghana. Five specific clusters of CDS were also found to be exclusively present with the African ST88 genomes (Fig. 2C). These CDS were
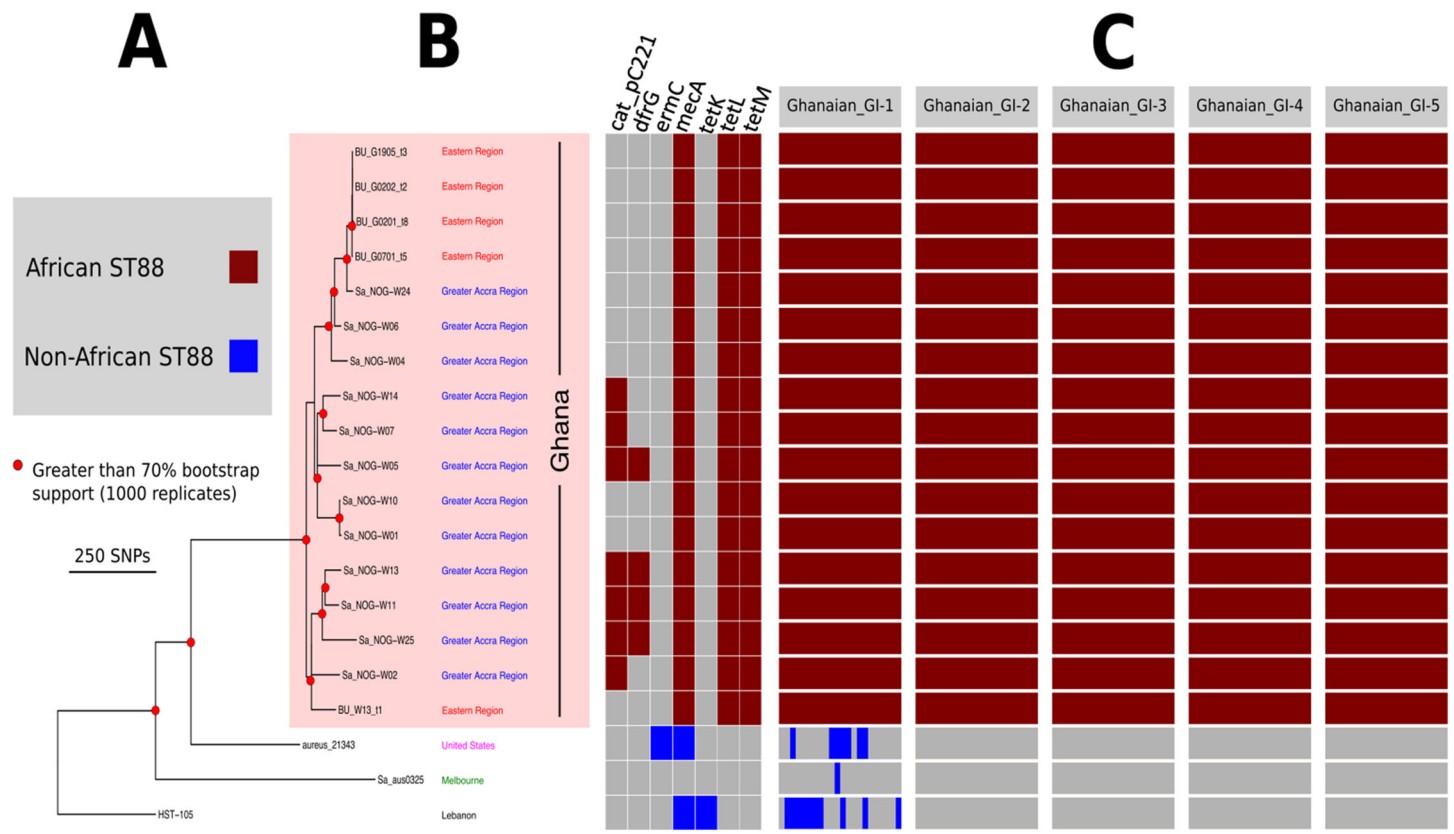

**Figure 2** **High resolution ST88 phylogeny and accessory genome analysis.** (A) Phylogeny inferred by read-mapping and variant identification among only ST88 genomes. Tree was produced using FastTree based on a pairwise alignment of 1,759 non-recombinogenic core genome SNPs among the 20 ST88 genomes. All major nodes in the tree (red circles) had greater than 70% bootstrap support (1,000 replicates). (B) Accessory gene content variation among the 20 ST88 genomes as assessed by ortholog comparisons using Roary. (C) Distinct genomic islands (GI) identified in Ghanaian isolates.

different to the five genomic regions identified in all ST88 relative to other *S. aureus* clones (Fig. 1B) suggesting that they were horizontally acquired by an ST88 MRCA that has since spread in Ghana (although a significantly larger sampling effort would be required to confirm this hypothesis). These regions harbour CDS suggestive of plasmid, phage and other mobile DNA elements (Table S2). We conducted a phylogeographic analysis to formally assess the relationship between the Ghanaian ST88 phylogeny and the specific geographic origin of the isolates, based on patient villages. However, there was no correlation between geography and phylogeny, suggesting again that the spread of ST88 in Ghana has been recent and rapid (Fig. 3).

## Phenotypic and genotypic antibiotic resistance

All 17 Ghanaian ST88 isolates harboured a SCC*mec*-IV [2B] cassette, and displayed phenotypic resistance to *β*-lactams, tetracycline and chloramphenicol (Table 1). Isolates were variably resistant to erythromycin, clindamycin, trimethoprim, amikacin and streptomycin (Table 1). There was agreement between phenotypic and inferred genotypic resistance (Fig. 1B). For the four genes (*blaZ, mecA, tetL* and *tetM*) detected in all 12 ST88 isolates from the Greater Accra Region, resistance correlated with phenotypic resistance to

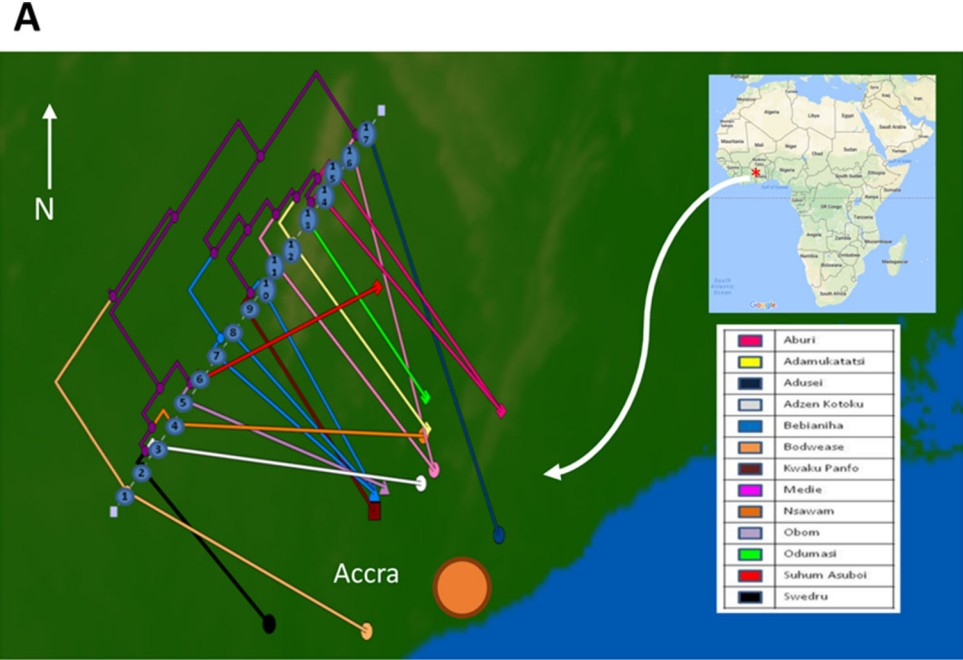

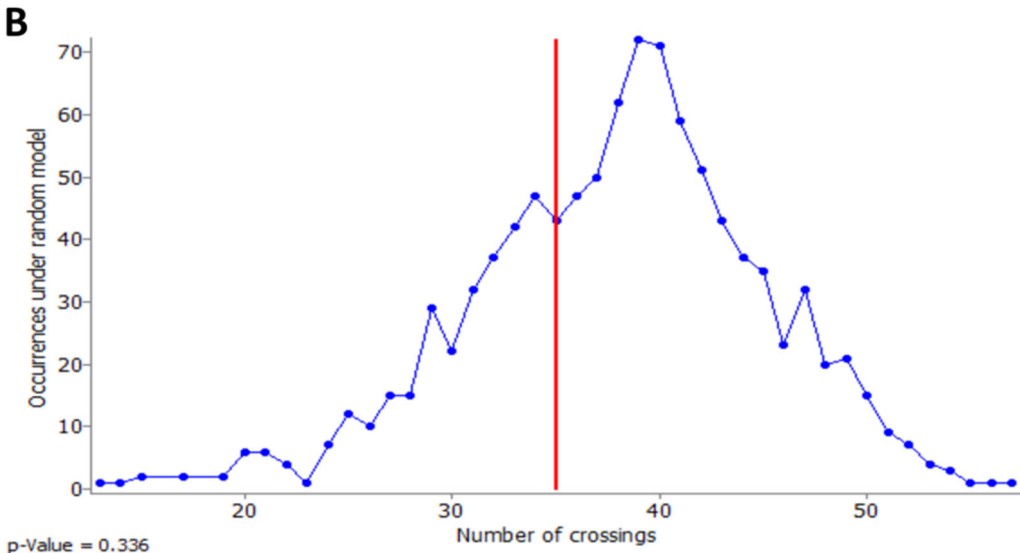

p-Value = 0.336

**Figure 3 Relationship between phylogeny of Ghanaian ST88 and their geographic origin.** (A) Phylogeographic alignment of phylogeny against isolate origin geography performed with GenGIS software and (B) Monte-Carlo analysis following 1,000 permutations of tree tips and geography of originating villages. The arrangement derived from the data was not significantly different to that which is expected by chance alone ($p > 0.05$), indicating a lack of geographical structure among the ST88 genomes.

all $\beta$-lactams and tetracyclines. Six isolates showed phenotypic and genotypic resistance to chloramphenicol (Table 1; Fig. 1B). Five of these isolates were from the same health centre; however, the time of isolation and the geographic origins of the patients were different, suggesting that these isolates are widespread across the region and were not acquired from a common source.

## CONCLUSION

The analysis presented here suggests that *S. aureus* ST88-IV is an emerging CA-MRSA clone in Ghana. This has the potential to become a serious public health threat, with implications for the treatment of *S. aureus* infections in Ghana, where there is no developed surveillance infrastructure to monitor antibiotic resistance. The abuse and misuse of antibiotics by health care givers and patients in Ghana are extensive (*Kpeli et al., 2016*). The development of resistance is furthermore encouraged by the widespread availability of higher classes of antibiotics to lower level health centres from regional medical stores, in addition to the unrestricted sale of these medicines to over-the-counter medicine sellers by pharmaceutical wholesalers, even though existing laws are supposed to limit the scope of these facilities to handle such medicines. Also implicated and widely documented are the prescribing practices of clinicians; with over-reliance on presumptive treatment and haphazardly prescribing antibiotics without recourse to due laboratory investigation. CA-MRSA has undergone rapid evolution and expansion worldwide. Because of its epidemic potential and limited treatment options, vigilance and antibiotic stewardship programmes need to be put in place to prevent further spread.

## ACKNOWLEDGEMENTS

We are grateful to the healthcare workers and patients at the Ga West and Ga South Municipalities and to Miss Nana Ama Amissah for giving us access to genome data.

### Funding

This work was supported by the Stop Buruli initiative of the UBS Optimus foundation and the Volkswagen Foundation. The funders had no role in study design, data collection and analysis, decision to publish, or preparation of the manuscript.

### Grant Disclosures

The following grant information was disclosed by the authors:
Stop Buruli initiative of the UBS Optimus foundation and the Volkswagen Foundation.

### Competing Interests

Timothy Stinear is an Academic Editor for PeerJ.

### Author Contributions

- Grace Kpeli conceived and designed the experiments, performed the experiments, analyzed the data, wrote the paper, prepared figures and/or tables and reviewed drafts of the paper.
- Andrew H. Buultjens conceived and designed the experiments, performed the experiments, analyzed the data, wrote the paper, prepared figures and/or tables and reviewed drafts of the paper.

- Stefano Giulieri performed the experiments, analyzed the data, contributed reagents/materials/analysis tools, wrote the paper and reviewed drafts of the paper.
- Evelyn Owusu-Mireku performed the experiments, reviewed drafts of the paper.
- Samuel Y. Aboagye performed the experiments, reviewed drafts of the paper.
- Sarah L. Baines analyzed the data, reviewed drafts of the paper.
- Torsten Seemann analyzed the data, contributed reagents/materials/analysis tools and reviewed drafts of the paper.
- Dieter Bulach analyzed the data, contributed reagents/materials/analysis tools and reviewed drafts of the paper.
- Anders Gonçalves da Silva contributed reagents/materials/analysis tools and reviewed drafts of the paper.
- Ian R. Monk analyzed the data, wrote the paper and reviewed drafts of the paper.
- Benjamin P. Howden reviewed drafts of the paper.
- Gerd Pluschke contributed reagents/materials/analysis tools and reviewed drafts of the paper.
- Dorothy Yeboah-Manu conceived and designed the experiments, contributed reagents/materials/analysis tools and reviewed drafts of the paper, supervision.
- Timothy Stinear conceived and designed the experiments, performed the experiments, analyzed the data, contributed reagents/materials/analysis tools, wrote the paper, prepared figures and/or tables and reviewed drafts of the paper, supervision.

### Ethics

The following information was supplied relating to ethical approvals (i.e., approving body and any reference numbers):

Ethical clearance was obtained from the institutional review board of the Noguchi Memorial Institute for Medical Research (NMIMR) (Federal-wide Assurance number FWA00001824). All study participants were well informed of the study objectives and written informed consent was obtained either from the patient or from the guardian of the patient.

### DNA Deposition

The following information was supplied regarding the deposition of DNA sequences:

1. The *S. aureus* ST88 raw sequence reads are accessible via ENA, Project PRJEB15428 (http://www.ebi.ac.uk/ena/data/view/PRJEB15428).
2. The AUS0325 Chromosome Assembly is accessible via ENA, LT615218 (http://www.ebi.ac.uk/ena/data/view/LT615218).
3. The AUS0325 PacBio raw sequence reads is accessible via ENA, ERS1354601 (http://www.ebi.ac.uk/ena/data/view/ERS1354601).
4. The MiSeq Data (Raw) for SA_NOG sequences described here are accessible via GenBank accession numbers: ERR1638070 to ERR1638081.

## Data Deposition

Strain data protocol has been deposited in Figshare: DOI 10.6084/m9.figshare.3863475.

## Supplemental Information

Supplemental information for this article can be found online at http://dx.doi.org/10.7717/peerj.3047#supplemental-information.

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
