# Peer review of "Genomic analysis of ST88 community-acquired methicillin resistant Staphylococcus aureus in Ghana"

_PeerJ, doi:10.7717/peerj.3047_

## Round 0.1 · original submission · Minor Revisions

· Academic Editor

Minor Revisions

Dear Dr Kpeli,

As you can see from the comments of the two expert reviewers, your manuscript was well perceived, and both reviewers found the results very interesting.

However, they indicated several minor issues you should all address before resubmitting a revised version of the article.

For further instructions on the resubmission process, please look at the detailed informations given below.

Kind regards,
Elisabeth Grohmann

Reviewer 1 ·

Basic reporting

1. Table 1. Abbreviations “spa, agr and PVL” might not be known by readers outside the staphylococci community. Please write out in full in the table legend.
2. Table 2. What is meant by “-1” and “-2” in “ST88-1” and “ST88-2”. Please explain.
3. Table 3. Please include origin of the isolates (e.g. Africa, Lebanon) and a column specifying CA-MRSA, HA-MRSA etc.
4. Figure 1+2. Please improve resolution of both figures with respect to resistance gene names and nucleotide positions (Fig 1A only)
5. Fig S1/Fig S2. Please add figure description
6. Please provide accession numbers for raw read data

Experimental design

no comments

Validity of the findings

1. lines 158ff. Please include a short description and interpretation of the CDS located on GI-5.
2. lines 176ff. The clusters of CDS found in the Ghanaian isolates are apparently different to those present in ST88 of other countries (nomenclature ST88_GI-1 and Ghanaian GI-1). Coincidentally, 5 genomic islands were described in both cases which might cause confusion by the reader (see also line 177: “as discussed…”). Please clearly state that these regions are different and are not interchangeable. Also, please include another table as supplementary information (Table S2) describing the CDS encoded by Ghanaian GI-1 to GI-5.

Comments for the author

The study very well describes the detailed molecular investigation of a particular CA-MRSA lineage based on whole genome sequencing using state-of-the-art techniques and precise bioinformatics analyses of the data. Results are robust, consistent, conclusive in interpretation and informative.

Reviewer 2 ·

Basic reporting

The authors present their results regarding the emergence of CA-MRSA in Ghana. They employed whole genome sequencing for ST88 CA-MRSA isolates and compared the results with other MRSA lineages worldwide.
The manuscript is written well and the knowledge gap that is being addressed by the authors is clearly stated. The Introduction gives relevant background information. Overall, the structure conforms PeerJ standard. However, the tables could need a bit more explanation, e.g. the genotypes, table 1, were not discussed and relevance of these were not made clear. Overall design of table 3 is not consistent. The figures are helpful to understand the experimental results; however resolution sometime makes it difficult to read the text in the figures, especially in printed form (e.g. Fig 1B aac_6_aph…; Fig 2C).

Experimental design

The experimental design is well thought and the description of the methods used is accurate. The research question is defined, relevant and meaningful.
However, the origin of the reference ST88 strain AUS0325 has not been described well. Also, references for mecA PCR are missing.

Validity of the findings

The findings and data the authors present are robust with some speculation. The overall conclusion is well stated and meaningful.

Comments for the author

Lines 54-55: do the cited references also refer to strains ST8-USA, ST93, ST1, ST80, ST59?
Line 58: Reference missing
Lines 60-61, Lines 66-67: ST8-IV[2B], ST-88 [2B] or ST-IV, ST-88?
Line 90, 155: GenBank
Line 91-92: 12 isolates from 11 patients; thus 1 patient is source of two different isolates? Do these correlate?
Line 9:8 Reference for mecA PCR is missing
Line 113-119: Is it Nullarbor or Nullabor??
Line 116: FriPan
Line 132: Could the authors explain a bit more in detail why AUS0325 was chosen and add a reference for this strain
Line 133: Tn552 (552 in italics)
Table 1
Define abbreviations spa, agr, PVL
Lines 2-4: sometimes “-“, sometimes “=” is used
Table 3
NC_007793.1
Inconsistent usage of sub sp vs subsp, also sometimes with or without “.”
“aureus” in italics

---

## Round 0.2 · accepted · Accept

· Academic Editor

Accept

Dear Grace,

Both reviewers and myself are in favour of publishing the revised version of your manuscript. You addressed and implemented the reviewer´s comments very well.

Congratulations!

Best regards,
Elisabeth

Elisabeth Grohmann
Academic Editor of PeerJ

Reviewer 1 ·

Basic reporting

no comment

Experimental design

no comment

Validity of the findings

no comment

Comments for the author

The authors very well addressed and implemented the reviewers‘ comments.

Reviewer 2 ·

Basic reporting

no further comments

Experimental design

no further comments

Validity of the findings

no further comments

Comments for the author

no further comments